# Too Much Dietary Flexibility May Hinder, Not Help: Could More Specific Targets for Daily Food Intake Distribution Promote Glycemic Management among Youth with Type 1 Diabetes?

**DOI:** 10.3390/nu14040824

**Published:** 2022-02-16

**Authors:** Angelica Cristello Sarteau, Elizabeth Mayer-Davis

**Affiliations:** 1Department of Nutrition, University of North Carolina at Chapel Hill, 245 Rosenau Drive, Chapel Hill, NC 27599, USA; mayerdav@email.unc.edu; 2School of Medicine, University of North Carolina at Chapel Hill, 245 Rosenau Drive, Chapel Hill, NC 27599, USA

**Keywords:** type 1 diabetes, adolescent, child, diet, food, nutrition, glycemic control, diabetes management

## Abstract

Average glycemic levels among youth with type 1 diabetes (T1D) have worsened in some parts of the world over the past decade despite simultaneous increased uptake of diabetes technology, thereby highlighting the persistent need to identify effective behavioral strategies to manage glycemia during this life stage. Nutrition is fundamental to T1D management. We reviewed the evidence base of eating strategies tested to date to improve glycemic levels among youth with T1D in order to identify promising directions for future research. No eating strategy tested among youth with T1D since the advent of flexible insulin regimens—including widely promoted carbohydrate counting and low glycemic index strategies—is robustly supported by the existing evidence base, which is characterized by few prospective studies, small study sample sizes, and lack of replication of results due to marked differences in study design or eating strategy tested. Further, focus on macronutrients or food groups without consideration of food intake distribution throughout the day or day-to-day consistency may partially underlie the lack of glycemic benefits observed in studies to date. Increased attention paid to these factors by future observational and experimental studies may facilitate identification of behavioral targets that increase glycemic predictability and management among youth with T1D.

## 1. Introduction

People living with type 1 diabetes (T1D) have 2–6-fold the mortality risk of the general population, are approximately 10-fold more likely to develop cardiovascular disease, and are disproportionately burdened by other costly physical and psychological complications that also manifest at a younger age than in people without T1D [1,2,3,4]. Meeting glycemic management targets (HbA1c < 7%) as soon after disease onset and as consistently as possible throughout the life span is therefore critical to minimize adverse health outcomes [5,6]. As demonstrated by the Diabetes Control and Complications Trial Research Group, even a few years of sub-optimal glycemic levels increase morbidity and mortality risks over 30 years later [7,8,9].

Glycemic management is more likely to be sub-optimal in adolescence through young adulthood than at other points during the lifecycle due to a convergence of physiological and behavioral factors [10,11,12,13,14]. Only ~14–17% of youth achieve HbA1c targets in settings with the most advanced diabetes technology [14,15]. Counterintuitively, average HbA1c in youth with T1D worsened in the US over the past decade despite concurrent increased uptake of self-management technology, underscoring the continued need for additional strategies to support glycemic management in this population [16,17,18].

## 2. Where Are the Evidence-Based Eating Strategies for Glycemic Management in Youth with T1D?

To summarize the state of the evidence regarding eating strategies that promote glycemic management among youth with T1D, in October 2021 (updated in January 2022), we employed the Cochrane rapid review strategy to search without date limitations combinations of terms describing young people with TID and eating behavior (see search strategy in Table 1 and Figure 1) in MEDLINE and Scopus, yielding 1224 records [19]. After duplicates were removed (*n* = 31), articles were screened and excluded (*n* = 982) if the abstract indicated the publication was unrelated to T1D, individuals 5–24 years of age (i.e., age range encompassing children, teenagers, and youth according to the United Nations), glycemic management, eating behavior, or an outdated version of a publication (i.e., consensus guidelines). Full-text articles were assessed for eligibility (*n* = 211), which identified 17 experimental studies that included eating behavior as an exposure and glycemic management as an outcome, 31 observational studies that included adjusted associations between eating behavior and glycemic management, 7 relevant systematic reviews and meta-analyses, as well as 4 consensus guidelines.

Despite widespread acknowledgement of the importance of nutrition for glycemic management in T1D, there is a paucity of evidence-based eating strategies, especially for youth [8,20,21,22]. Contrary to popular perception, rigorous testing of eating strategies to support glycemic management is extremely limited, inclusive of the carbohydrate counting and low glycemic index (low-GI) strategies that are most widely counseled in clinical practice and promoted by published consensus guidelines [23,24,25,26]:

Only three randomized clinical trials have been conducted to test the glycemic effects of carbohydrate counting as compared to a control group or alternative eating strategy among youth with T1D [27,28,29]. Gilbertson et al. tested a carbohydrate counting diet compared to a more flexible low-GI diet among youth 8–13 years of age (carbohydrate counting *n* = 49, low GI *n* = 55) and found that those in the low-GI diet group—not the carbohydrate counting group—had lower HbA1c after one year [27]. Spiegel et al. tested intensive carbohydrate counting education among youth 12–18 years of age (intervention *n* = 33, control *n* = 33) with no difference in HbA1c or carbohydrate counting accuracy observed between control and intervention groups at 3–4 months [29]. Although one clinic-based observational study found 102 youth 8–18 years of age were able to estimate the carbohydrate content of 73% of 17 meals and snacks within 10–15 g of the actual amount, the other studies on this topic describe low carbohydrate counting accuracy in youth and challenges to improve accuracy among youth and adults alike [29,30,31,32,33,34,35,36]. The aggregate available data about effectiveness and feasibility of carbohydrate counting in youth with T1D call into question the universal appropriateness of and principal reliance on this eating strategy to manage glycemia at this life stage. Indeed, the two experimental studies (18 month pre–post trial among 25 youth 7–15 years of age by Marigliano et al. and 24 month randomized control trial among 84 youth 7–18 years of age by Göksen et al.) that tested quarterly carbohydrate counting education among youth and observed modest reductions in HbA1c implemented the strategy in conjunction with specific food options for meals or in addition to an existing prescribed meal plan [28,37]. Additionally, growing evidence about the non-negligible glycemic impacts of protein and fat as well as intra- and inter-individual variability in glycemic response to combinations of macronutrients or types of food, further cast doubt upon the ability of the carbohydrate counting—even if perfectly mastered—to optimize glycemia when used as the predominant eating strategy [38,39]. Indeed, while some trials in adults have found carbohydrate counting can result in improved glycemia, systematic reviews and meta-analyses show no overall benefit for glycemic management and other health indicators [40,41,42,43].

Observational investigations of GI and glycemic levels include a secondary analysis by Nansel et al. with 3 day dietary records collected among 252 youth 8–18 years of age at three time points during an 18 month trial as well as a study by Crouse et al. that examined eating episodes from three 24 h dietary recalls together with overlapping postprandial blood glucose and HbA1c data among 87 youth 12–17 years of age [44,45]. Neither study found an association between GI and glycemic levels. Six small (*n* = 16–23), short-duration (1–5 days) crossover feeding trials have been conducted to test the effect of a low-GI diet on glycemic management in youth with T1D [46,47,48,49,50,51]. These controlled feeding trials investigated the glycemic effects of one low-GI macronutrient-matched meal compared to one high-GI macronutrient-matched meal, or the effects of alternating days of low-GI meals and usual meals, which demonstrated lower postprandial glycemia and daytime mean glycemia. However, besides the aforementioned 12 month study conducted by Gilbertson et al., only one trial of relatively longer duration and in a real-world setting has been conducted in young people with T1D to test the glycemic effects of a low-GI diet. In a 3 month parallel-group pilot trial by Marquard et al., 17 youth 6–14 years of age followed an optimized mixed diet (i.e., a German analogue to a diet developed from the American Healthy Eating Index) or a low-GI diet [52]. No difference in HbA1c was observed between groups. Study authors speculated whether focusing more on overall pattern of eating instead of rules around individual foods or macronutrients (i.e., carbohydrate counting and low-GI approaches) may be more beneficial. This proposition has since been reiterated by a growing number of researchers in light of poor diet quality and stagnating improvement or worsening of average glycemic management over time in youth with T1D [44,53,54,55,56,57,58,59,60,61,62,63].

However, to date, there are only three experimental studies that have examined the glycemic benefits of an overall pattern of eating among youth with T1D. Although one observational study conducted in the SEARCH for Diabetes in Youth Cohort (*n* = 2130, 10–22 years of age) found that greater adherence to the Dietary Approaches to Stop Hypertension (DASH) diet was associated with a slightly more optimal HbA1c (0.2% lower in highest vs. lowest tertile), Peairs et al. found that 3 day adherence to a personalized DASH diet resulted in no difference in glycemic variability as compared to usual intake [64,65]. Although there was a suggestion of improvement in secondary outcomes (average blood glucose and time spent in hyperglycemia), the study was an underpowered, three-day controlled feeding trial in 16 youth 11–17 years of age, limiting confidence in efficacy as well as understanding of generalizability, real world feasibility, or long term effects of this approach. In a single-arm, pre–post trial among 96 pediatric clinic patients with T1D, Cadario et al. tested the effect of Mediterranean diet counseling on lipid profile and HbA1c at 6 months, but did not observe a change in HbA1c [66]. Nansel et al. conducted a randomized control trial among youth 8–16 years of age and their families (intervention *n* = 66, control *n* = 70), which tested the effect of a behavioral intervention to increase whole-plant food intake on co-primary outcomes dietary quality and glycemic management, but there was no difference in HbA1c observed between groups at 18 months [67].

Observational analyses focused on overall pattern of eating have repeatedly pointed to positive associations between fat intake and HbA1c, as well as higher intake of carbohydrates and sub-optimal glycemic management [61,68,69,70,71,72,73,74,75,76,77,78]. Ultimately, different sample distributions of nutrient intake and covariates as well as dissimilar variable specification and model adjustment decisions across available studies, coupled with a dearth of rigorous experimental testing of different overall patterns of macronutrient intake, have together challenged the translation of the observational evidence into guidance that differs markedly from that recommended to the general population [24].

Our review of the literature suggests that no eating strategy tested to date among youth with T1D is robustly supported by the existing evidence base, which is characterized by few experimental studies, small study sample sizes, and lack of replication of study results due to marked differences in experimental design and eating strategy tested.

## 3. Is It “Time” to Consider Daily Food Intake Distribution Targets?

There is agreement across clinical guidelines that “matching insulin to food” is fundamental for glycemic management in T1D [24,42,79]. Operationally, this requires concordance between timing, type, amount, and frequency of food consumed with timing, type, amount, and frequency of exogenous insulin administered throughout the day. International Society for Pediatric and Adolescent Diabetes (ISPAD) and Diabetes Canada guidelines include statements about the potential benefit of day-to-day consistency in food intake routine and attention to timing and spacing of food throughout the day [24,79]. However, no eating strategies tested since the advent of flexible insulin regimens have combined behavioral targets for food timing, type, amount, and frequency. That the strategies examined in real-world settings in the past two decades have focused on macronutrients or food types without incorporating timing, frequency, or day-to-day consistency in food distribution may partially underlie the lack of glycemic benefits observed in experimental studies to date, as well as the mixed results (i.e., null, positive, and negative associations) noted in observational analyses between different overall patterns of nutrients or foods and glycemic management [55,68,70,78,80,81]. A limited, mainly observational evidence base lends credence to this hypothesis: European and US studies of youth with T1D describe a high prevalence of “grazing” or “skipping” food distribution patterns or pattern inconsistency across multiple days as well as associations between these eating behaviors and sub-optimal HbA1c [80,82,83,84,85].

The Norwegian Childhood Diabetes and Quality Project found that skipping meals was associated with higher odds of sub-optimal HbA1c (>7.5%) among 655 youth with T1D (mean age: 11.4 years) [86]. Another study from the project further demonstrated among 550 youth 2–19 years of age that even when intensive insulin treatment with insulin pumps and multiple injections were used, not skipping meals and having breakfast and supper regularly (defined as six times per week) was significantly associated with improved HbA1c [82]. Other observational studies among youth with T1D have suggested that more consistent breakfast consumption and not skipping lunch are associated with more optimal glycemic management, and that children who are offered food ad libitum instead of through an eating pattern defined by set meals are more likely to have sub-optimal HbA1c [61,80,87,88].

The importance of eating frequency for glycemic management has been highlighted by a handful of observational analyses [59,80,84]. Among 821 T1D youth in the SEARCH for Diabetes in Youth cohort, eating frequency (≤3, 4–5, or 6–10 times/day) measured at baseline and follow-up visits was significantly related to HbA1c measured repeatedly over 5 years [84]. Increased eating frequency was associated with larger increases in HbA1c: for those who ate ≤3 times per day at the outset and ate 6–10 times per day 5 years later, longitudinal models predicted greater absolute increases in HbA1c (2.77%). Despite the observation of an association in the opposite direction among Type 1 Diabetes Exchange youth (*n* = 262)—each additional eating occasion was associated with a decrease in HbA1c of 0.11%—results from this cross-sectional analysis still reinforce the same message that eating frequency matters in glycemic management.

Although there is a paucity of published studies examining day-to-day consistency in eating practices among individuals with T1D, Wolever et al. reported day-to-day variation of carbohydrate, but not fat or protein, to be positively related to HbA1c in 272 youth with T1D, which remained significant when adjusted for age, sex, duration of diabetes, and BMI (*p* = 0.0097) [85]. Monzon et al. also demonstrated through a 72 h observational study among 39 children that postprandial glycemic variability was greater for those who ate more carbohydrate and protein than typical for themselves during lunch—a meal that was also defined by large variability in time of day it was consumed as compared to other meals [70].

In aggregate, the above summarized observational evidence base, albeit limited, illustrates that food distributed throughout the day in a grazing or skipping pattern and day-to-day pattern variability are associated with HbA1c, thereby highlighting the potential glycemic benefit of targeting daily food intake distribution and consistency in addition to food group and macronutrient intake that have been the focus of the eating strategies tested to date.

Further underscoring the potential benefit of focusing on daily food intake distribution and consistency, a growing number of small and short-term studies in adults without T1D demonstrate a range of metabolic benefits of placing eating occasions and certain food groups within specific time windows throughout the day, limiting daily intake to a total number of hours, as well as maintaining day-to-day consistency in food intake distribution [89]. These strategies are increasingly understood to be beneficial because they synchronize eating with the circadian regulation of metabolic hormones and processes [89]. Of particular relevance to T1D, insulin sensitivity decreases and postprandial glucose responses increase as the day progresses, which may contribute to difficulty predicting and dosing insulin for the postprandial response to the same food eaten at different times of day [41,90,91]. Eating when hormones that counter insulin are elevated may also lower predictability of glycemic response: melatonin levels are elevated shortly after waking and again in the late evening, which inhibits insulin action and may require that food be dosed with a greater amount of exogenous insulin than if it were consumed at a different time of day [91,92,93,94,95].

Acknowledgement that time of day affects glycemic response to food in T1D is reflected in common use of higher insulin to carbohydrate ratios in the morning as compared to the afternoon [96]. That insulin needs vary for an eating occasion based on whether it was preceded by a smaller or larger eating occasion also highlights how daily inconsistency in food timing, amount, and composition of eating occasions could more easily result in inaccurate insulin dosing and sub-optimal glycemic levels [96]. Given that multiple factors converge to influence glycemia throughout the day and the limitations of insulin dosing strategies to account for all of these factors and their interactions with each other, establishing approximate consistency around timing and frequency of eating occasions in addition to amount and type of food eaten at each eating occasion may increase predictability of glycemic response to food throughout the day, make insulin dosing calculations easier and more accurate, and improve overall glycemic management. Importantly, this strategy may improve glycemic management throughout the day not only by reducing insulin dosing errors that can result from high variability in eating behavior (i.e., timing, frequency, sequence, food composition and amount), but also because the structure introduced by establishing a consistent routine around daily food intake distribution may result in establishing greater consistency in other factors that have immediate and latent effects on glycemic levels, such as exercise and sleep [97,98,99,100].

Some preliminary observational evidence supports the potential benefit of this strategy among youth with T1D. Cross-sectional analyses have illustrated that the proportion of a pediatric T1D clinic population in Australia who met glycemic targets rose from 31% in 2004 to 83% in 2016, during which time an intensive diabetes management program was implemented that included individualized guidance for minimum and maximum carbohydrate quantity at a consistent number of meals and snacks consumed at approximately the same time each day [101]. When associations were examined between adherence to this guidance and time in range in an independent sample of matched continuous glucose monitoring and 24 h dietary recall data from 109 American youth (mean age 14.8 years), eating behavior that followed more as opposed to less of this guidance throughout the day was associated with approximately 9% more time in range overall, including 14% more time in range at night (*p* = 0.02) [102].

An eating strategy with targets for timing, frequency, and amount eaten across a sequence of meals and snacks, and approximate day-to-day consistency of this food intake distribution, may also benefit glycemic levels, not to mention overall cardiovascular risk, by promoting healthy weight. Concurrent with the rise in average HbA1c in recent years, overweight among youth with T1D in Australia, Europe, and the United States has risen to surpass the prevalence in the general population [16,73,103]. Overweight challenges glycemic levels through disruptions in insulin sensitivity as well as fat and carbohydrate metabolism, while also compounding the already excess cardiovascular risk among individuals with T1D via overweight-associated risk factors (e.g., hypertension, dyslipidemia, inflammation, and oxidative stress) [104,105]. The previously described study from the Norwegian Childhood Diabetes and Quality Project among 655 youth with T1D (mean age 11.4 years) found that skipping meals was not only associated with sub-optimal HbA1c, but also associated with higher odds of overweight (OR: 2.8, *p* = 0.03) [83]. Associations between a skipping or grazing food intake distribution and overweight have also been repeatedly shown by observational studies among adolescent populations without T1D [106,107].

In contrast, short-duration (1 week–4 months) studies testing eating strategies with time-based eating behavior targets, such as limiting eating to a consistent window of time throughout the day, have repeatedly shown mean improvements in body weight and fat percentage in healthy adults (*n* = 10 studies) and adults with overweight, prediabetes, or diabetes (*n* = 9 studies) [89,108]. Increasing evidence also suggests the cardiovascular benefits (e.g., reduced blood pressure, triglycerides, and insulin resistance) may also be independent of weight loss in humans—a finding that has been repeatedly demonstrated in animal studies [108]. That daily consistency in an overall time window for food intake may improve metabolic health independent of weight loss suggests that an eating strategy that incorporates this behavioral target may benefit both overweight and non-overweight individuals with T1D. Further, because studies demonstrate insulin intensification leads to weight gain in T1D, time-based eating behavior targets may not only help treat but also prevent insulin resistance [8,109].

## 4. Considerations and Future Directions

The dearth of literature examining strategies that emphasize a consistent daily food intake distribution with approximate eating behavior targets for timing, frequency, and food amount across a sequence of meals and snacks may reflect reaction against the restrictiveness of eating plans under fixed insulin regimens and the perception that youth and their parents would not adhere to these behavioral targets. However, as with all guidance implemented in real-world settings, food intake distribution targets derived from empirical evidence could be individualized to create a food intake distribution that consistently fits within the routines of youth and their families and incorporates usual preferred foods. Such targets could include a range instead of precise numbers (i.e., time window for each meal and snack, and range of grams eaten at a meal or snack) to accommodate inevitable fluctuations in schedule that occur as part of daily life. Targets could be adjusted over time as schedules and activities change more substantively and permanently. Further, depending on youth and family resources, abilities, and preferences, past and future advances in insulin dosing strategies, including use of exchanges, carbohydrate counting, and technological assistance (e.g., phone applications) could be incorporated to support food flexibility within the food amount and composition targets for each meal and snack [110,111,112]. As part of the development process of a larger study, we conducted a small six-week feasibility pilot of an eating strategy defined by five behavioral targets for food intake distribution that was iteratively individualized for youth (*n* = 9) and their families’ routines, abilities, and preferences [102]. Reports by youth and parents gave preliminary indication of high perceived acceptability and sustainability of this strategy [102].

Almost all observational and experimental studies to date concerned with eating patterns and glycemic management in youth with T1D have focused on macronutrients or food groups, with a few focused on usual frequency of eating but without incorporating information about timing or content of each eating occasion [27,59,67,80,88]. Although analytical constraints of questionnaires largely underlie these decisions (e.g., food frequency questionnaires), interestingly, this analytical focus has persisted even when multi-day weighted dietary records or 24 h dietary recalls with granular frequency, timing, and food amount information sampled at more than one time point are available [29,44,45,52,58,59,61,67,68,74,80,88,113,114,115].

In order to facilitate the specification of food intake distribution targets to experimentally test among youth with T1D, future research must both elucidate the glycemic benefits that stem from day-to-day consistency in food intake distribution as opposed to the benefits that stem from different daily food intake distributions (i.e., different combinations of meal or snack timing, frequency, sequence, and food amount). Such research would require, at minimum, consecutive days of dietary data sampled over an extended period of time, information about food timing, type, and amount at each eating occasion, and matched glycemic data from same-day continuous glucose monitoring as well as long-term glycemic management estimates either from HbA1c measurements or aggregate continuous glucose monitoring measures. Information about amount and timing of insulin administered matched with food intake distribution data across multiple days could also not only help tease out the extent to which the glycemic effects of certain food intake distribution patterns may be explained by insulin behavior, but also shed light upon the extent to which different food intake distribution patterns may facilitate more accurate estimation of insulin needs throughout the day.

In addition to limited collection or adjustment for confounding factors (e.g., insulin dosing behaviors, disordered eating, blood glucose monitoring, treatment of hypoglycemia, and socioeconomic factors), lack of standardization in how meals and snacks are defined (e.g., defined by participant, time window, calorie content, percent of total energy, and percent of total calories from specific macronutrients) are also a notable weakness in the observational evidence base that has focused on food intake distribution and glycemic management in T1D, which inhibits the utility of observational studies to inform the design of experimental studies [116,117,118]. Replication of adjusted observational analyses using the same definition for eating occasions would be a start in making sense of food intake distribution patterns that may be beneficial for glycemic management and developing targets to test in experimental studies. These studies should be conducted across levels of diabetes technology engagement and other resources, so as to generate insights about the interplay of eating behavior, glycemic management, and emerging technological tools (e.g., closed-loop systems) while also ensuring that the eating behavior guidelines they inform are applicable to more than a few subgroups of youth with T1D –a population whose diabetes self-management resources vary widely around the world.

Ultimately, prospective studies are not only needed to test the efficacy of consistent daily food intake distribution targets for glycemic management among youth with T1D, but also acceptability to youth and their families—evidence-based eating strategies are only as effective as they are feasible for and valuable to the people they are designed to serve. Such assessments will better facilitate the development of strategies that target potentially modifiable factors of acceptability (e.g., neophobia and pickiness) and enable modification of or cessation of investment in strategies when obstacles to acceptability are less readily modifiable (e.g., incompatibility with local culture or resources) [115,119]. There is much to learn about whether reducing some freedom in food-related decisions by specifying targets and consistency related to food intake distribution will increase freedom via more predictable and manageable glycemia among youth with T1D.

## Figures and Tables

**Figure 1 nutrients-14-00824-f001:**
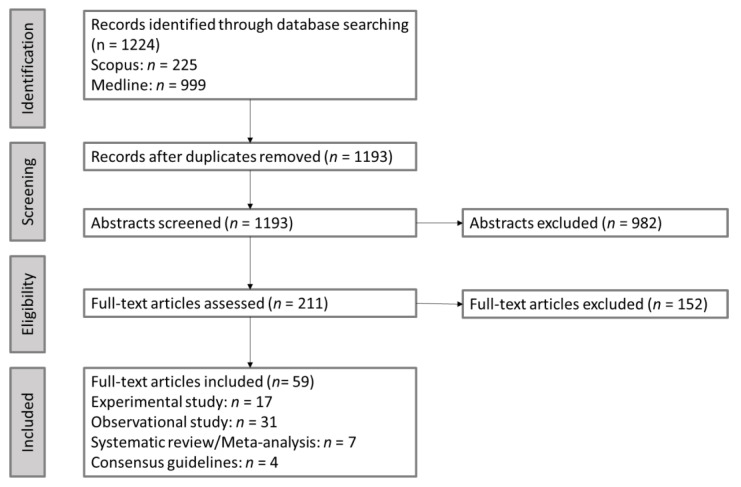
PRISMA Flow Diagram.

**Table 1 nutrients-14-00824-t001:** Search Strategy Keywords.

Population		Eating Behavior
children or young or adolescent or teen or youth	type 1 diabetes or type 1 diabetes mellitus or type 1 or T1D or T1DM or DM1	nutrition or diet or food or eating or eating strategy or eating frequency or food frequency or food timing or food time or mealtime or food distribution or carbohydrate or fat or protein or energy or meal or snack

## Data Availability

Not applicable.

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
