# Peer review of "Too Much Dietary Flexibility May Hinder, Not Help: Could More Specific Targets for Daily Food Intake Distribution Promote Glycemic Management among Youth with Type 1 Diabetes?"

_nutrients, 2022, doi:10.3390/nu14040824_

Round 1

Reviewer 1 Report

The authors provided a descriptive review on how the glycemic control in adolescents could be improved by different dietary strategies. There is no doubt that this issue is very important since the handling of adolescents with T1D is extremely complicated and the glycemic control of this population should be improved. Unfortunately this personal view of the authors does not convince the reader which startegy is better and does not add significant information to our present knowledge.

Under the "Considerations and future directions" subtitle the authors mention their own feasability pilot study. It would be much mor exciting to hear about these results.

The methodology of literature search is not mentioned and we do not get information how the authors selected from the available studies. Were there selection critera? Although, the contribution of ACS and EMD authors was methodology, no methodology is part of the manuscript.

Author Response

We thank the reviewer for the helpful review and comments. We have since included the methodology that we used as part of our narrative review, which was selected according to rapid review guidance put forth by Cochrane.1

We respectfully disagree with the reviewers view that our manuscript does not add significant information to present knowledge. Based on our systematic literature review, there is no publication that synthesizes the current state of the evidence base of eating strategies that have been examined among youth with type 1 diabetes. In this way, we contribute to the broader field’s homing in on effective and feasible eating strategies that will enable youth to better manage their diabetes.

We also thank the reviewer for the interest in our research! Our feasibility pilot study is just now undergoing recruitment and will not be completed until early 2023. We look forward to publishing them as soon as possible to contribute learnings to this much needed area of study.

1. Garritty, Chantelle, et al. "Cochrane Rapid Reviews Methods Group offers evidence-informed guidance to conduct rapid reviews." Journal of clinical epidemiology 130 (2021): 13-22. 

Reviewer 2 Report

This is an interesting, well written and helpful review. Focusing on type 1 diabetes eating strategies in young people with type 1 diabetes is a little debated topic that raises suggestive and unresolved questions in terms of clinical counseling: How useful is dietary flexibility in achieving good metabolic control?

I would suggest highlighting a couple of concepts that might make it even more compelling to consider flexibility in nutrition counseling for children with diabetes.

1-Focusing only on the quantity of macronutrients, and more precisely on carbohydrates, is not enough as can be seen for example from the variation of the ISPAD guidelines which in 2018 suggested a reduced quantity of carbohydrates compared to 2014 and from a recent observational study that reports the association between a better time-in-range with an even lower carbohydrate share (doi: 10.3390 / nu13113869).

2-Furthermore, the widespread use of advanced technologies in diabetes, such as closed-loop control, facilitates the administration of insulin at meals and could lead to greater flexibility of the diet. Further studies are needed to better understand the relationship between food flexibility and the use of advanced technologies. I would return to this concept in the last paragraph of the document.

Author Response

We thank the reviewer for the thoughtful review and helpful suggestions.

We greatly appreciate the suggested reference to the Cherubini et al. publication, which was published just after our systematic literature review was completed in October 2021. We have since added the reference to help illustrate the portion of the manuscript that addresses this topic.

We thank the reviewer for the suggestion to highlight the importance of research that generates insights about the relationship between eating behavior and technology to ensure applicability of eating recommendations to a broad range of youth with type 1 diabetes given the increasingly widespread use of tools that offer the possibility of greater dietary flexibility. We have since added this point to the conclusion section (second to last paragraph).

Reviewer 3 Report

Many compliments to the authors for their brilliant review. I really appreciated the stress of the evidence on the importance of nutrition in managing type 1 diabetes despite technological advances. It’s obvious to me, but I know it’s not that obvious to every pediatric diabetologist.

I believe that many journals dealing with pediatric diabetes have a certain difficulty in accepting documents that indicate a tendency to food "restrictions" and "rules" compared to those that describe technological advances in diabetology.

Only few comments:

  • In the abstract too often mentions the word "management". Authors try to find synonyms
  • Delete between the key words "disease management" and replace with "diabetes management"
  • In Italy we conducted an interesting study that tried to suggest a proper management of the pizza meal, considered a forbidden meal for patients with type 1 diabetes mellitus. In addition to the control of pizza preparation (recipe and leavening times) CHO-based dosage, insulin action time, as well as interference with other meals or exercise, have been meticulously controlled. The result was an excellent control of that meal with a simple wave bolus from insulin pump. I think it may be a good suggestion for what the authors mean in the conclusions (lines 226-230). This is the link (doi: 10.1089/dia.2019.0191. Epub 2019 Aug 8. PMID: 31335171).

4)  Line 52: At the end of the sentence there are ":" rather than "."

Author Response

We thank the reviewer for the compliments, thoughtful review, and helpful revisions.

We have incorporated the suggested text edits. We have also varied the verbiage in the abstract to avoid repetition of the word “manage” or “management”.

We thank the reviewer for the suggested reference. We agree that it illustrates a useful, nuanced example of research advances in the accuracy of insulin dosing strategies that could facilitate flexibility in food routines so that preferred or usual foods (i.e., pizza) can be healthfully incorporated into the eating patterns of youth with type 1 diabetes. We have since added the citation to our submission.

Round 2

Reviewer 1 Report

I suggested rejection. Since the manuscript's scientific soundness has not improved, therefore I do not want to change my opinion. However, I have to admit, that this is my personal impression and if other reviewers have different views I accept and appreciate them.